# Precision Medicine and Novel Therapeutic Strategies in Detection and Treatment of Cancer: Highlights from the 58th IACR Annual Conference

**DOI:** 10.3390/cancers14246213

**Published:** 2022-12-16

**Authors:** Sean P. Kennedy, Oliver Treacy, Emma H. Allott, Alex J. Eustace, Niamh Lynam-Lennon, Niamh Buckley, Tracy Robson

**Affiliations:** 1School of Biological, Health and Sports Sciences, Technological University Dublin, D07 ADY7 Dublin, Ireland; 2Discipline of Pharmacology and Therapeutics, College of Medicine, Nursing and Health Sciences, University of Galway, H91 TK33 Galway, Ireland; 3Patrick G. Johnston Centre for Cancer Research, Queen’s University Belfast, Belfast BT9 7AE, UK; 4Department of Histopathology and Morbid Anatomy, Trinity Translational Medicine Institute, Trinity College Dublin, D08 HD53 Dublin, Ireland; 5National Institute for Cellular Biotechnology, Dublin City University, D09 NR58 Dublin, Ireland; 6Department of Surgery, Trinity St James’s Cancer Institute, Trinity Translational Medicine Institute, St James’s Hospital, Trinity College Dublin, D02 PN40 Dublin, Ireland; 7School of Pharmacy, Queen’s University Belfast, 97 Lisburn Road, Belfast BT9 7AE, UK; 8School of Pharmacy and Biomolecular Sciences, Royal College of Surgeons Ireland, D02 YN77 Dublin, Ireland

**Keywords:** targeted therapeutics, early detection, epigenetics, cancer vaccine, systems biology

## Abstract

**Simple Summary:**

The Irish Association for Cancer Research (IACR) held its 58th annual conference from the 30th of March to the 1st of April 2022, in Cork, Ireland. The following article is a report of knowledge conveyed at the conference. There was a focus on cancer prevention in both “Early Detection” and “Cancer Vaccines” sessions, and cancer treatment in “Novel Therapeutics” and “Breakthrough Research” sessions. The research presented at this conference highlighted the interplay of both prevention and treatment, as many speakers focused on the same cancer but at different times in the treatment process. There was a push for more non-invasive early detection techniques, the current impact of cancer vaccines and new ways of further stratifying selection criteria for easier identification of high-risk individuals. This was coupled with novel treatment strategies and identification of new therapeutic targets.

**Abstract:**

Innovation in both detection and treatment of cancer is necessary for the constant improvement in therapeutic strategies, especially in patients with novel or resistant variants of cancer. Cancer mortality rates have declined by almost 30% since 1991, however, depending on the cancer type, acquired resistance can occur to varying degrees. To combat this, researchers are looking towards advancing our understanding of cancer biology, in order to inform early detection, and guide novel therapeutic approaches. Through combination of these approaches, it is believed that a more complete and thorough intervention on cancer can be achieved. Here, we will discuss the advances and approaches in both detection and treatment of cancer, presented at the 58th Irish Association for Cancer Research (IACR) annual conference.

## 1. Introduction

Cancer is a disease that has a considerable impact on society and on the individual. Currently, 1 in 2 men and 1 in 3 women will be diagnosed with cancer within their lifetime. In Ireland, the incidence of cancer is over 45,000 cases per annum (www.cancer.ie (accessed on 1 July 2022) and is expected to double again by 2050. Not only does this disease have a huge impact on the individual, but it has the most devastating economic impact of any cause of death in the world (www.cancer.org (accessed 1 July 2022)). The global economic impact of cancer was $895 billion dollars in 2008. This is 19% higher than heart disease, which has the second largest impact at $753 billion (www.cancer.org (accessed 1 July 2022)).

The ability to both accurately target and treat cancer has been the benchmark for success in the cancer research community, since the introduction of targeted therapeutics by Sidney Farber in 1948 [1]. With Sidney Farber’s seminal discovery of anti-cancer agents, it was said that “*only through the cooperation of both prevention and treatment, will it be possible to produce a significant reduction in cancer mortality*”. In the past 30 years, there has been a 32% drop in the cancer mortality rate, which has been attributed to, among other things, improvements in prevention and treatment (www.cancer.org (accessed 1 July 2022)). At the 58th Irish Association of Cancer Research (IACR) conference, the sessions were reflective of this mantra, in that we had speakers looking towards early detection of cancer, preventative “cancer vaccines” and treatment through both mechanistic understanding of pathways as well as targeting them through novel therapeutics.

## 2. Early Detection and Diagnosis

Like two sides of the same coin, prevention and treatment go hand in hand. The ultimate goal of a health care professional is to prevent a disease from taking hold and requiring treatment. In Prof. Sam Janes group at University College London (UCL) they pose the question “Could there be a future that we prescribe a cancer preventative, when people quit smoking?” When adjusted for both age and sex, global lung cancer mortality (18%) is almost double the 2nd highest mortality cancer, colorectal (9.4%) [2], however when it comes to treatment costs, in the case of non-small cell lung cancer (NSCLC), the vast majority of money is spent on treatment of high grade tumours, carrying lower survival rates, and not on the detection and prevention of the disease [3].

Prof. Janes echoes remarks made by Peter B Bach, MD of Memorial Sloan Kettering Cancer Centre (MSK), that cost of therapy in established, high grade cancers are soaring on a logarithmic scale and unsustainable [4]. To combat increasing costs in later stages, Prof. Janes’ lab look to eliminate those costs through detection at earlier stages. A way of determining whether pre-invasive lung carcinoma-in-situ (CIS) lesions progress to invasive cancer or regress to low grade dysplasia [5] is highlighted by Prof. Janes. The difference is observed through upregulation of genomic instability CIN genes (ACTL6A, ELAVL1, MAD2L1, NEK2, OIP5) in their ability to predict progression through pre invasive CIS, however CIS lesions are similar to cancer on a molecular level [5].

This led Prof. Janes to his next question, “What are the immune characteristics of these pre-cancerous lesions?”. Evidence of different evolutionary trajectories in gene expression for early precancerous legions in tumour evasion of the immune system [6], as well as both progressive and regressive CIS lesions with both immune “hot and cold” phenotypes, threaten to muddy the waters [7]. Prof. Janes then went on to show that regressive and progressive legions can be successfully stratified via chromosomal instability which predicts progressive legions, while immune competence predicts regressive legions [7]. In a more translational approach, Prof. Janes then goes on to describe the EARL study (NCT03870152) which is the first randomised trial of lung cancer prevention, testing electrocautery versus surveillance of histologically confirmed high-grade pre-invasive airway lesions. This collaborative trial between Cancer Research UK and the lung cancer centre of excellence at UCL is still recruiting and it aims to reduce cancer mortality in the same way other trials have successfully done through low dose computed topography (LDCT) [8].

Finally, Prof. Janes talked about the SUMMIT study (NCT03934866) in which the objectives were twofold; to assess the feasibility of LDCT screening of a large metropolitan population and to assess the accuracy of a blood biomarker for multiple cancers. Despite road blocks to success like COVID-19 and historically poor uptake and retention of patients in lung screening trials in general [9], the SUMMIT study has recently reported high patient satisfaction with the study [10].

## 3. Systems Biology Approach

When trying to tease apart complex biological systems and illuminate non-canonical pathways involved in cancer progression, systems biology is a tool which can deliver significant insight into otherwise opaque systems. In the Irish Cancer Society lecture and selected talks session, Prof. Chris Sweeney highlighted a systems biology approach towards uncovering an unknown regulator of Nuclear Factor-Kappa B (NF-κB) and its involvement in “lethal” prostate cancer. Both canonical and non-canonical NF-κB and its involvement in cancer have been described and interrogated at length over the last 2 decades [11,12].

Prof. Sweeney began the talk highlighting early ventures into understanding the mechanism behind NF-κB being present in early prostate cancer but expression having no clear association with “lethal” prostate cancer [13]. While it is clear that NF-κB is activated and overexpressed in multiple types of prostate cancer [13], it was shown that there is an increased nuclear localisation in cancer cells [14] and correlation of NF-κB immunoreactivity with disease recurrence [15], highlighting that if increased NF-κB is nearly always present in early prostate cancer, then what determines whether NF-κB drives “lethal” prostate cancer? To address this question, Prof. Sweeney utilised a systems biology approach in defining a key regulator of NF-κB activation in “lethal” prostate cancer, namely Tristetraprolin (TTP) (also called ZFP36).

NF-κB activation is controlled by both positive and negative feedback mechanisms which leads, inevitably, to multiple layers of complexity [16]. Through a collaboration with Prof. Travis Gerke’s lab and a bioinformatical approach to the question posed earlier, it was elucidated that low TTP expression is associated with “lethal” prostate cancer [17]. This conclusion was reached by reconstructing specific NF-κB activation pathways in the context of “lethal” prostate cancer through 3 steps. Firstly, the data was analysed heterogeneously (that is that the NF-κB pathway was defined in a large and diverse dataset), the data was then configured via a Bayesian, biology-specific integration and then finally, differential gene profiles were queried with a clinical context. This led to the finding that TPP is a putative prostate cancer tumour suppressor gene which regulates NF-κB via increased mRNA degradation of the NF-κB activator (TNF-α) and prevention of the nuclear translocation of the p65 subunit of NF-κB [17].

Prof. Sweeney showed currently unpublished work using both the cancer genome atlas (TCGA) RNAseq and Dana Farber Cancer Institute (DFCI) IHC data that TPP loss decreases disease-free survival in primary human prostate cancer and compounds impact of PTEN loss in localised prostate cancer by ~40%. To observe the effects of TPP loss on prostate cancer progression alone and in combination with PTEN loss, a genetically modified mouse model was developed recapitulating that phenotype. The result was acceleration of prostate cancer progression, driven by PTEN loss which was also reflected in human clinical correlation data. TPP loss facilitated increased tumour cell proliferation, invasion and dissemination, as well as quicker progression to castration-resistant prostate cancer.

In the final part of Prof. Sweeney’s presentation, he discussed the use of Dimeththylaminoparthenolide (DMAPT) as a way of targeting NF-κB, highlighting its excellent toxicity profile, bioavailability and notably a protectant quality against radiation and cisplatin in normal tissue, while conversely a sensitizer in tumours [18]. Despite NF-κB having a range of pathway “on and off ramps”, it is a viable and druggable target in prostate cancer. DMAPT is a unique anti-cancer agent in that it displays a preferential effect on cancer cells over non-cancer cells, due to cancer cells having an aberrant glutathione balance [19,20] and a greater reliance for survival on constitutive NF-κB activation [11,12,16]. Prof. Sweeney went further and showed how DMAPT limits radiation-induced apoptosis in non-cancer but increases in cancer tissue and prevents radiation-induced mitochondrial damage in prostate and bladder cancer [18]. Prof. Sweeney concluded his presentation with a whole host of unpublished, in vivo model data (lung, urothelial, bladder, kidney and prostate) in which he demonstrated in vivo radiation efficacy against cancer cells and protection for normal cells. Preliminary human pharmacokinetic data showed a modest suppression of NF-κB in patients with acute myeloid leukemia using a limited dose of DMAPT. This promising result paves the way for work on an encapsulated formulation and clinical trials thereafter.

## 4. Cancer Vaccines

Cancer Vaccines are constructed with the goal of inducing an immune response against tumour antigens [21]. Despite great success with both Human papilloma virus (HPV) [22] and Hepatitis B virus (HBV) [23], there have been constraints which dictate the success of the vaccine approach. Heterogeneity of a tumour has long been a limiting factor when it comes to treatment, as it allows for multiple different avenues of therapeutic resistance and as cancer vaccination can be raised to a single epitope and can’t account for various neoantigens simultaneously, this problem persists. To combat this heterogeneity, some groups have looked at targeting heterogeneity with individual, personalised vaccines, however, that brings with it a whole host of logistical and regulatory problems [24]. Prof. Benoit Van den Eynde of the Ludwig Institute for Cancer Research (LCR) has been looking at tackling heterogeneity through targeting MAGE-type antigens in lung cancer.

Prof. Van den Eynde showed that there have been many attempts so far to develop MAGE-targeting cancer vaccines. Some very large clinical trials involving thousands of patients have been carried out, aimed at evaluating a recombinant MAGE-A3 protein vaccine and adjuvant formulations in NSCLC and melanoma patients bearing MAGE-A3 expressing tumours. However, both have unfortunately failed (NCT00480025) (NCT00796445). With advances in oncology, it is hypothesized that the reasons for these failures arise from suboptimal induction of CD8+ T cells and the vaccine not utilized in combination with checkpoint inhibitors. To rectify this, Prof. Van den Eynde proposed the use of a heterologous prime-boost (a Chimpanzee Adenovirus Oxford (ChAdOx) recombinant, in combination with a Modified Vaccinia Ankara (MVA)) viral vector, which has been shown to induce a strong CD8+ T cell response in humans [25].

The reason MAGE-A3 is such an attractive vaccination target is because MAGE-A3 is the most frequently expressed MAGE-type antigen in human cancers, as well as the established oncogenic role of many MAGE-type antigens [26,27]. It is also necessary to target NY-ESO-1 in order to induce a strong CD8+ response which was missing from the unsuccessful trials. Taking on lessons from the past, Prof. Van Den Eynde then spoke of a clinical trial project in which he can use a vaccine to “heat the cold tumours” and sensitise them to anti-PD-1 (pro-T cell). This approach in which they combine the ChAdOx1/MVA/MAGE-A3/NY-ESO-1 vaccine with the new standard of care, namely chemotherapy (Carboplatin and Paclitaxel) and anti-PD1 (pembrolizumab), in MAGE-A3/NY-ESO-1 tumours was performed in DBA/2 mice with 15V4T3 (NSCLC) cell-derived tumours. After optimisation of treatment schedules and exposure to the vaccine, a significant benefit of the vaccine/anti-PD-1/chemo combination was observed, when compared to any other combination of each component [28]. This has led to the development of a collaborative (VOLT) Phase 1/2a clinical trial on NSCLC in which inclusion criteria are MAGE-A3 +/− NY-ESO-1 expression and the two arms are separated into vaccine/standard of care (SoC) against SoC alone. The implications of a functional, interchangeable cancer vaccine which promotes CD8+ T cell infiltration are huge in terms of personalised medicine.

## 5. Cancer Epigenetics

Formerly seen as a genetic disease, cancer has been shown to be a disease of many hats, including epigenetic abnormalities [29]. In recent years, many avenues of epigenetics have been investigated, including non-coding mRNA expression, DNA methylation and in the case of Prof. Adrian Bracken’s lab at Trinity College Dublin (TCD), histone modifications. Prof. Bracken began the talk showing recent rounds of NGS sequencing and the key oncogenes and tumour suppressors highlighted [30] and a reminder that chromatin-modifying proteins are one of the main contributors to that list.

Two different groups applied genome-scale sequencing to pediatric diffuse intrinsic pontine gliomas (DIPG) and found that this deadly brain cancer was caused by a frequent (80%) somatic mutation in Histone H3 and Lysine 27 (more commonly written H3K27me#) [31,32]. Prof. Bracken then shows how Polycomb Repressive Complexes 2 (PRC2) mediated H3K27me3 recruits PRC1 to compact chromatin. EZH2 (a component of PRC2) is recurrently mutated in lymphomas with increased H3K27me3 and decreased H3K27me2 [33]. It was shown previously that core PRC2 is also deleted in various cancers such as T cell leukemia, myeloid malignancies, and malignant peripheral nerve sheath tumours [34].

To address these destructive epigenetic mutations in EZH2, Prof. Bracken’s lab has aided in the development of highly selective EZH2 inhibitors. Utilizing a high throughput screening process for inhibitors of the PRC2 complex, followed by pharmacodynamic and pharmacokinetic optimization, low nanomolar potency small molecule EZH2 inhibitors were generated [35]. These small molecules displayed remarkable selectivity for EZH2 and showed similar efficacy against wild-type and mutant forms of EZH2. One of these therapeutics, Tazemetostat, produced clinically meaningful and durable responses, has a favourable safety profile in heavily pretreated patients with follicular lymphoma and has FDA-granted, accelerated approval for adult patients with relapsed or refractory follicular lymphoma whose tumors are positive for an EZH2 mutation [36].

To understand how H3K27M2/3 disrupts the epigenome and trial these new small molecule inhibitors, Prof. Bracken’s lab created a clinically-relevant model of DIPG [37]. Inhibition of EZH2 enzymatic activity through Tazemetostat was shown to selectively reverse aberrant repression of neurodevelopmental genes associated with DIPG [37]. Prof. Bracken then surmised the talk with a thought on both understanding and effectively targeting the right H3K27 aberrant phenotype, either through EZH2 or Histone deacetylase inhibitors (HDAC) in aberrant repression associated with elevated H3K27me3 expression or reduced gene expression associated with elevated H3K27ac expression [38].

## 6. Novel Therapeutic Strategies

This year’s winner of the European Association of Cancer Research (EACR) senior investigator prize was Dr. Daniela Ottaviani of the Royal College of Surgeons Ireland (RCSI) and their work on CDK12 as a novel therapeutic target in endocrine therapy-resistant breast cancer was presented. Despite being one of the most studied forms of cancer, breast cancer still eludes researchers with resistance channels to therapeutics. Out of 2.3 million women diagnosed with breast cancer in 2020, 73% of cases will be estrogen receptor positive (ER+) and human epidermal growth factor receptor 2 negative (HER2-) and of this 73% treated, a staggering 40% of ER+/HER2- patients will develop resistance to endocrine therapy [39]. Endocrine resistance represents a major clinical problem and often leads to secondary metastatic disease so to combat this, strategies to overcome such resistance and identify new actionable targets are needed [40]. Because of its prevalence in advanced metastatic breast cancer [41] and association with worse survival outcome in ER+ breast cancer [42], Dr. Ottaviani elected to investigate the functional and mechanistic role of cyclin-dependent kinase 12 (CDK12) in endocrine resistant metastatic breast cancer, and evaluate its use as a new therapeutic target.

Through whole exome sequencing, Dr. Ottaviani showed CDK12 amplification is a frequent and common alteration found in advanced breast cancer (*n* = 78). When utilising RNA sequencing, CDK12 gene expression was also shown to be increased in metastasis vs. primary breast tumours (*n* = 45). This was further confirmed through liquid chromatography—mass spectrometry (LC-MS) of ER+ primary breast tumours with good and poor outcomes and high CDK12 protein expression were shown in ER+ primary tumours with poor outcome (*n* = 10). Pathway analysis revealed CDK12-driven transcriptome in endocrine resistance drives ER+ signalling, contributing to cancer cell growth, cell cycle progression and proliferation. Upon looking at transcription factors and chromatin regulators potentially cooperating with CDK12, MED1 stood out as a potential target for its known involvement with CDK12 in tumourigenesis [43]. MED1 was seen as a potential transcriptional partner of CDK12 as ligand-bound ER recruits the MED1/Mediator complex to initiate targeted gene transcription [44]. Dr. Ottaviani went on to show how MED1 and ER are depleted from the chromatin upon siCDK12, indicating CDK12 knockdown disrupts MED1 and ER recruitment to the chromatin. Following CDK12 knockdown, apoptosis was significantly increased as evidenced by both RNA sequencing and PARP cleavage western blots. CT7116 (a CDK12 inhibitor) (300 nM) was then shown to significantly inhibit viability and clonogenic growth of ER+ cell and organoid models of advanced breast cancer.

Dr. Ottaviani showed CDK12 facilitates endocrine resistance in ER+ breast tumours by promoting estrogen-dependent growth signalling through modulation of MED1 and ER chromatin accessibility and then displayed pharmacological targeting of CDK12 against viability of cell and organoid models of ER+ advanced breast cancer. These preliminary results support the potential use of CDK12 inhibition in the treatment of endocrine therapy-resistant breast cancer.

## 7. Conclusions

Prevention and treatment are akin to diplomacy and war. When diplomacy fails, war becomes inevitable. In that same way, both prevention and treatment are important avenues of attack and are highlighted at the 58th IACR annual conference. This conference couldn’t have been better concluded than with the final speaker and recipient of the IACR Award for Outstanding Contribution to Cancer Medicine and Research, Prof. John Reynolds. The academic head of clinical surgery at St. James Hospital and Trinity College Dublin (TCD), Prof. Reynolds is also the national lead in Ireland for oesophageal and gastric cancer. He is the president of the Irish Society of Clinical Nutrition and European Society of Diseases of the Oesophagus, so it was poignant when he spoke about one of the more deceptively preventable diseases, malnutrition.

Prof. Reynolds highlighted the famous Lancet article linking body-mass index and incidence of cancer [45], which identified Oesophageal adenocarcinoma as the cancer with the highest incidence risk per 5 kg/m^2^ bodyweight. Throughout the talk, Prof. Reynolds championed taking the initiative with terms such as “prehabilitation” and “preoperative” when talking about nutrition-related carcinomas and talked about a gap in the Irish dietetic services which leaves little opportunity for early intervention or effective ongoing support. To finish on a positive note, Prof. Reynolds highlighted the ICBP SURVMARK-2 study, displaying Ireland’s 5-year net survival of both Oesophageal and Stomach cancer and how survival has increased by 11% in both cancers over the last 20 years [46] due in no small part to preventative and early intervention.

## Data Availability

Not applicable.

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
