# Peer review of "Precision Medicine and Novel Therapeutic Strategies in Detection and Treatment of Cancer: Highlights from the 58th IACR Annual Conference"

_cancers, 2022, doi:10.3390/cancers14246213_

Round 1
Reviewer 1 Report
This report by Kennedy et al. provides a very nice and comprehensive overview of the topics discussed at the 58th Annual IACR Conference. The authors concisely summarise the latest research advances regarding early detection and prevention of cancer, as well as novel treatment approaches, presented by renown speakers at the conference. The report is well written and provides a good summary for the current state of the field. I therefore recommend accepting this report for publication in Cancers.
Author Response
Dear Sir/Madam,
Thank you so much for your kind words and swift review of our manuscript. I will submit it to the editor as per your recommendation.
Many thanks again,
Sean
Reviewer 2 Report
The manuscript highlights some presentations in the 58th IACR annual conference. Although some results from the related presentations are discussed in the text, lacking of the details might affect the significance of the paper and readers’ interest.
Author Response
Dear Sir/Madam,
Many thanks for your reply and swiftness of your review. The inherit brevity of the article does unfortunately make it hard to encapsulate the significance of some of the researchers findings however I tried to highlight the key findings and include as many references to the original research as possible.
Many thanks again for your time and input.
Sean